# Effects of BMSC-Derived EVs on Bone Metabolism

**DOI:** 10.3390/pharmaceutics14051012

**Published:** 2022-05-08

**Authors:** Xuchang Zhou, Hong Cao, Jianming Guo, Yu Yuan, Guoxin Ni

**Affiliations:** 1School of Sport Medicine and Rehabilitation, Beijing Sport University, Beijing 100084, China; mdzxsj@163.com; 2School of Kinesiology, Shanghai University of Sport, Shanghai 200438, China; chhcaohong@163.com (H.C.); gjm911239@126.com (J.G.); yuanyumail@126.com (Y.Y.)

**Keywords:** extracellular vesicles, exosomes, bone formation, bone resorption, bone metabolism, BMSCs, cell therapy

## Abstract

Extracellular vesicles (EVs) are small membrane vesicles that can be secreted by most cells. EVs can be released into the extracellular environment through exocytosis, transporting endogenous cargo (proteins, lipids, RNAs, etc.) to target cells and thereby triggering the release of these biomolecules and participating in various physiological and pathological processes. Among them, EVs derived from bone marrow mesenchymal stem cells (BMSC-EVs) have similar therapeutic effects to BMSCs, including repairing damaged tissues, inhibiting macrophage polarization and promoting angiogenesis. In addition, BMSC-EVs, as efficient and feasible natural nanocarriers for drug delivery, have the advantages of low immunogenicity, no ethical controversy, good stability and easy storage, thus providing a promising therapeutic strategy for many diseases. In particular, BMSC-EVs show great potential in the treatment of bone metabolic diseases. This article reviews the mechanism of BMSC-EVs in bone formation and bone resorption, which provides new insights for future research on therapeutic strategies for bone metabolic diseases.

## 1. Introduction

Bone is a dynamic organ that undergoes modeling and remodeling throughout life in response to various physiological and pathological stimuli, especially mechanical stress stimuli [1]. Under normal conditions, bone resorption and bone formation are performed, respectively, by osteoclasts and osteoblasts, whose activity is orchestrated by the osteocyte, which maintains the other bone cells in a dynamic balance [2]. Bone marrow mesenchymal stem cells (BMSCs), also known as bone marrow mesenchymal stromal cells, are important precursors of osteoblastic-lineage cells [3]. During the process of bone remodeling, triggered by osteocytes [2], the old or damaged bone is cleared by osteoclasts, while growth factors are released from the bone matrix, which subsequently induces the recruitment and migration of BMSCs to specific parts and eventually leads to the differentiation of BMSCs into osteoblasts and the formation of new bone in specific parts [4,5]. However, under pathological conditions, such as estrogen deficiency, abnormal mechanical stress and drug side effects, the dynamic stability of bone formation and bone resorption is disrupted, resulting in osteoporosis (OP), fractures or incomplete healing of bone defects [6]. Therefore, the modulation of coupled bone formation and bone resorption is a fundamental strategy for the prevention and treatment of bone metabolism-related diseases.

BMSCs are pluripotent stem cells with self-renewal, immune regulation and multi-directional differentiation potential, which have the advantages of convenient access, low immunogenicity and the ability to secrete trophic factors [7,8]. As the main source of osteoblasts, BMSCs are important contributors to the process of bone tissue repair [9,10]. It has been reported that BMSCs not only replace and repair damaged tissue through cell proliferation and differentiation directly but also promote damaged tissue repair through paracrine action indirectly [11,12]. Therefore, BMSC-based cell therapy is considered to have great potential in the treatment of bone metabolism-related diseases [13]. However, there are still some unresolved problems in the clinical application of BMSC transplantation therapy, such as appropriate homing, cell dedifferentiation and low cell survival rate, as well as possible immune rejection and ethical issues [14]. Recent studies have shown that BMSCs may play a more important role in tissue regeneration and repair through paracrine function, rather than replacing damaged tissue directly [8,15]. Extracellular vesicles (EVs) are recognized as key mediators of intracellular signaling during the paracrine process of BMSCs [16]. BMSC-derived EVs (BMSC-EVs), with similar therapeutic efficacy and functional properties to their parental BMSCs, could evoke pro-regenerative effects similar to stem cell therapy by encapsulating and enriching cargo for delivery into damaged tissues [17,18,19]. EVs contain a large number of bioactive molecules such as proteins, lipids and nucleic acids (mRNA, ncRNA and DNA), which could effectively deliver intercellular messengers to adjacent cells or carry them to distant target cells via biological fluids, thereby modulating various physiological and pathological processes in recipient cells [20,21]. As a natural nanoscale bioactive molecular carrier, BMSC-EVs have unique advantages, such as higher stability, non-immunogenicity, intrinsic homing effect, non-tumorigenicity, non-ethical controversy, easier access and preservation, etc., compared with BMSCs [22,23]. It has very attractive therapeutic potential in various bone-related diseases including OP, fracture healing and bone defect repair [24,25,26]. In addition, the unique lipid bilayer of EVs ensures stable cargo delivery and protects biomolecules such as RNA and proteins from rapid degradation. More importantly, the BMSC-EVs cargo can be engineered and modified to improve the efficiency of bioactive molecules [27]. Therefore, modifiable BMSC-EVs may represent a safer and more promising therapeutic strategy. In recent years, BMSC-EVs have been gradually applied in bone disease research as a nanocarrier [28]. Numerous studies have shown that BMSC-EVs can promote osteogenesis, angiogenesis and bone mineralization, suggesting that BMSC-EV-based cell-free therapy is a promising strategy for bone regeneration and repair [29,30,31]. However, studies in bone metabolism that are based on the BMSC-EVs strategy are still in an early stage. An in-depth understanding of the specific mechanism of BMSC-EVs regulating bone regeneration will help to develop more effective therapeutic strategies. The current review, therefore, summarizes the specific mechanism of BMSC-EVs regulating bone metabolism and some existing issues and further provides a basis for future research of BMSC-EVs in bone metabolic diseases.

## 2. Overview of EVs

EVs are nanoscale phospholipid bilayer vesicles secreted by almost all cells [32]. After the fusion of multivesicular bodies with the cell membrane, EVs are released into the extracellular environment in the form of exocytosis and act as intercellular messengers to transfer their cargoes to homogeneous and heterogeneous recipient cells via ligand-receptor interactions, endocytosis or direct membrane fusion, thereby modulating their biological functions and altering the cellular microenvironment [33]. According to their diameter, composition and origin, EVs can be classified into three types: exosomes, microvesicles and apoptotic bodies [34]. Among them, exosomes are one of the most popular and attractive EV categories. Exosomes are vesicles with a diameter between 40 and 150 nm, formed by the invagination of intracellular lysosomal particles, and express markers such as cell adhesion molecules cluster of differentiation 9 (CD9), CD63, CD81, Hsp60 (heat-shock proteins 60), Hsp70 and Hsp90 [34,35]. Currently, the purification of exosomes is mainly based on the detection of exosome diameter and surface molecular markers [36]. The cell culture supernatant was taken through a series of centrifugation to remove dead cells and larger debris, and then the exosomes were obtained by ultracentrifugation to detect specific markers. In 1983, exosomes were first discovered in an in vitro experimental study of sheep reticulocytes [37]; however, the secretion of exosomes has long been mistaken as cellular “garbage” [38,39]. Until recent years, studies have found that exosomes act on various receptor cells through a variety of bioactive molecules in vesicles and play an important role in immune surveillance, angiogenesis, tumor development, metabolism and inflammatory responses [40]. To be more specific, exosomes have the following unique functions, which may provide new research ideas and therapeutic targets for the mechanistic pathways of various diseases. First, exosomes can bind two cells simultaneously through ligand interaction for information exchange without the need for direct cell-to-cell contact. Second, exosomes can bind to target cell membranes to enable cells to acquire new adhesion properties. Third, exosomes can also directly fuse with target cells to exert biological effects in cells through substance delivery [41]. Notably, although EVs have been classified, their precise nomenclature and specific markers that distinguish various vesicle subtypes are still lacking investigation [42]. The terms “EVs” and “Exos” are in fact not strictly differentiated, and both terms are used to refer vesicle species obtained by sequential centrifugation and filtration [43]. Since the concept of “EVs” includes “Exos”, in this article, we prefer to use “EVs”.

EVs are derived from a variety of cells, while EVs from different cell sources present heterogeneous characteristics and components [44]. In 1970, Friedenstein et al. [45] first identified the existence of MSCs. Subsequently, it was found that MSCs are widely distributed throughout the body, including bone marrow, adipose tissue, the umbilical cord and amniotic fluid, and play an indispensable role in bone metabolism, angiogenesis and immune regulation [46,47,48]. Due to the multi-directional differentiation potential of MSCs, including osteoblasts, chondrocytes and adipocytes, MSCs transplantation has been widely used in regenerative medicine [49]. At present, the main sources of stem cells used in bone regeneration and repair research include ucMSCs (umbilical cord blood mesenchymal stem cells), ADSCs (adipose-derived mesenchymal stem cells) and BMSCs [50]. Whether MSCs of different origins can be regarded as the same cell type remains controversial [3]. Studies have confirmed that ucMSCs derived from umbilical cord blood have weak osteogenic differentiation ability [51], and the role of ucMSC-derived exosomes (ucMSC-Exos) on bone formation is actually through promoting angiogenesis rather than osteogenesis [52]. BMSCs and ADSCs are relatively easy to obtain clinically, especially ADSCs, which are a promising source of MSC-Exos for bone regeneration. It must be admitted that, in practical use, researchers often choose MSCs based on availability and convenience. This results in the frequent selection of ADSCs that are abundant in adipose tissue for the extraction of EVs production [8]. However, BMSCs are the main source of osteoblasts and have significant osteogenic differentiation ability [44,53]. Previous studies have shown that BMSCs are more sensitive to osteoinductive culture conditions than ADSCs. Compared with ADSCs, BMSCs exhibit stronger osteoinductive potential [19]. Therefore, it is more reasonable to extract EVs from BMSCs for bone regeneration and repair [54,55].

## 3. miRNAs Mediating the Regulation of BMSC-EVs on Bone Metabolism

MicroRNAs (miRNAs) are evolutionarily conserved small non-coding RNAs with a length of about 22 nucleotides [56]. MiRNAs induce mRNA degradation or inhibit the expression of their target genes through complementary pairing with the base sequence of the 3’ UTR region of mRNA, thereby participating in biological processes such as cell differentiation, growth, migration and apoptosis [57]. miRNAs can be excreted not only into the local cellular microenvironment but also into the blood circulation, suggesting local and systemic effects [58]. However, single-stranded miRNAs secreted outside cells are unstable and easily degraded, limiting their applications [59]. Therefore, an efficient carrier is needed to transport and protect miRNAs. Numerous studies have shown that the transfer of BMSC-EVs that carries a variety of bioactive molecules to target cells affects their cellular functions and phenotypes through biochemical molecular crosstalk between cells [49,60]. Among the various biologically active molecules contained in exosomes, miRNAs are particularly attractive to researchers [61]. Natural carrier exosomes, as specialized membrane vesicles, can protect miRNAs from being degraded by RNases during transport and be endocytosed by recipient cells to regulate intercellular communication [62,63]. Exosomal miRNAs are stable in blood and may represent ideal biomarkers for various diseases [64]. miRNAs, as one of the important cargoes carried by BMSC-Exo, have been extensively studied due to their potential to promote or inhibit bone repair [11,65,66]. It is of great significance to explore the role of BMSC-Exo-packaged miRNAs (BMSC-Exo-miRNAs) in bone metabolic diseases. miRNAs have been reported to regulate multiple processes in bone, such as osteoblast and osteoclast proliferation and differentiation [67]. Recent studies have shown that exosomal miRNAs can regulate bone development [10], inhibit osteoclast activity [68] and improve fracture repair [26] by competitively inhibiting post-transcriptional gene expression. However, exosomes alone showed poor bone regeneration effects due to their limited osteogenic inductive ability [69,70,71]. Based on the internal cargos, the appropriate modification of BMSC-Exos contents to construct engineered exosomes through cellular engineering, such as the genetic modification or protein modification, provides an attractive strategy for enhancing the therapeutic effect of BMSC-Exos [72].

### 3.1. BMSC-EV-miRNAs Regulating Bone Formation through Spatiotemporal Dependence

The bone formation consists of three stages: proliferation, differentiation and matrix mineralization [73]. Osteoblasts, mainly derived from BMSCs with strong osteogenic differentiation potential, play an important role in bone formation [74]. Alkaline phosphatase (ALP) is a bone-specific enzyme that is considered a marker of early bone formation. The expression of ALP was positively correlated with the osteoblast differentiation level. The higher the activity of ALP, the more obvious the differentiation of pre-osteoblasts to mature osteoblasts [75]. Osteocalcin (OCN) is a specific protein of bone tissue. Most of the OCN synthesized and secreted by osteoblasts is deposited in the bone matrix and is considered to be a marker of the late stage of bone formation [76]. Furthermore, both runt-related transcription factor 2 (Runx2) and Osterix are common markers during osteogenic differentiation [77,78]. The elevated expression of Osterix promotes osteogenic differentiation, whereas the deficiency of it results in a lack of osteoblasts [79]. Many studies have shown that BMSC-Exos are closely related to the process of bone formation by transporting miRNAs, which is an attractive therapeutic strategy for bone metabolic diseases (as shown in Table 1). Li et al. [65] found that highly expressed miR-186 in BMSC-Exos could promote osteoblast proliferation through miRNA sequencing analysis. Further studies revealed that BMSC-Exo-miR-186 may enhance osteogenesis by directly targeting Mob1, a key gene that inhibits the Hippo signaling pathway [65]. Another study stated that the overexpression of miR-150-3p in BMSC-Exos upregulated the expression of osteogenic factors such as Runx2 and Osterix, promoted osteoblast proliferation and inhibited apoptosis. In contrast, the inhibition of miR-150-3p in BMSC-Exos showed the opposite effect [10]. It was shown that miR-150-3p in BMSC-Exos could promote the proliferation and differentiation of osteoblasts. Similarly, Zhang et al. [80] found that BMSC-Exos could deliver miR-935 into osteoblasts and subsequently inhibit the expression of its target gene signal transducer and activator of transcription 1 (STAT1), thereby promoting osteoblast proliferation and enhancing ALP activity and calcified nodules. STAT1 can interact with Runx2 in the cytoplasm to inhibit the nuclear localization of Runx2 [81]. The upregulation of STAT1 expression can significantly hinder the nuclear translocation of Runx2 to regulate osteoblast differentiation [82]. Therefore, miR-935-encapsulated BMSC-Exos may promote Runx2 nuclear localization by targeting STAT1 to enhance osteoblast proliferation and initiate osteogenic differentiation. The above studies suggest that the initiation of bone formation can be precisely regulated by appropriately modifying miRNAs encapsulated in BMSC-Exos. In addition, some studies have found that BMSC-Exo-miRNAs can also widely participate in the regulation of advanced osteogenic differentiation and matrix mineralization [11,83,84]. The miRNAs used in the above studies are all derived from BMSC-Exos, which turned out to be capable of differentially regulating the proliferation, differentiation and matrix mineralization stages of bone formation. Studies have reported that BMSC-Exos dose-dependently increased the activity of ALP and promoted the osteogenic differentiation and matrix mineralization of BMSCs [84,85]. Therefore, the above-mentioned differential regulation of osteogenic differentiation by BMSC-Exo-miRNAs might be due to the differences in the concentrations and doses of BMSC-Exos between different experiments. This suggests that the uniform and quantified dose of BMSC-Exos should be explored to improve the reproducibility of BMSC-Exos experiments and standardize the precise treatment of BMSC-Exos.

It is worth mentioning that BMSC-Exos extracted in the early stage of osteogenic differentiation can accelerate early osteogenic differentiation, and the exosomal cargo delivered by BMSCs in the later stage of osteogenic differentiation can stimulate extracellular matrix mineralization, which may be closely related to the changes in the expression profiles of miRNAs in BMSC-Exos [90]. Therefore, the extraction time of BMSC-Exos should also be focused on during the induction of osteogenic differentiation, as different cellular microenvironments may lead to differences in the contents of BMSC-Exos. This spatiotemporal difference is also consistent with age-related differences in the activity of BMSCs. Compared with young rats, BMSCs from aged rats exhibited significantly reduced stemness and osteogenic differentiation [91,92]. Xu et al. [63] demonstrated that BMSC-Exo-miR-31a-5p may be significantly increased with the aging of BMSCs and secreted into the extracellular microenvironment to influence osteoblast differentiation. Furthermore, compared with young BMSC-Exos, the level of miR-29a was significantly decreased in aged BMSC-Exos, implying that the decreased level of miR-29a in aged exosomes may lead to age-related bone loss, and miR-29a-loaded BMSC-Exos may be a potential treatment for age-related bone loss [9]. In summary, BMSC-Exo-miRNAs are involved in regulating bone formation through a spatiotemporal dependence manner.

### 3.2. Wnt/β-Catenin Signaling Mediating the Regulation of BMSC-EV-miRNAs on Bone Metabolism

Wnt/β-catenin signaling, as a conserved signaling pathway, is widely involved in cell proliferation, differentiation, migration and apoptosis and is especially related to the regulation of bone metabolism [93]. Zuo et al. [94] reported that BMSC-Exos attenuated radiation-induced bone loss and restored the balance between the adipogenic and osteogenic differentiation of BMSCs by activating Wnt/β-catenin signaling. However, this study did not delve into which key cargo within BMSC-Exos acts to inhibit bone loss. Previous studies have confirmed that miRNAs can promote the osteogenic differentiation of BMSCs through the Wnt/β-catenin signaling pathway [95]. Qin et al. [96] found that three osteogenesis-related miRNAs, miR-196a, miR-27a and miR-206, were highly enriched in BMSC-Exos by miRNA sequencing of BMSC-Exos. Subsequently, Peng et al. [88] found that, after the overexpression of miR-196a in human bone marrow mesenchymal stem cells (hBMSCs), miR-196a-enriched hBMSC-Exos could upregulate the protein expression of Wnt1 and β-catenin by targeting dickkopf-1(Dkk-1), a known negative regulator of the Wnt/β-catenin signaling pathway, upregulate the protein expression of Wnt1 and β-catenin and increase the expression of osteogenesis-related factors such as ALP, OCN and Runx2, thereby enhancing osteogenic differentiation. Conversely, when inhibiting miR-196a in BMSCs, the above expression was partially reversed, and the function of hBMSC-Exos to promote osteogenic differentiation was inhibited [88]. This indicates that miR-196a is an important functional active molecule in hBMSC-Exos. hBMSC-Exos can deliver miR-196a to osteoblasts by targeting Dkk1 to activate Wnt/β-catenin signaling to promote osteogenic differentiation. Dkk2, a member of the Dickkopfs family, can inhibit Wnt/β-catenin signaling, like Dkk1 [97]. Previous studies have shown that the expression of miR-27a is significantly increased during the process of the osteogenic differentiation of MSCs, while the expression of miR-27a is significantly decreased during the process of adipogenic differentiation, suggesting that miR-27a may be a key regulator of the differentiation fate of MSCs [98]. Wang et al. [66] reported that BMSC-EV-miR-27a alleviated bone loss in OP mice by ameliorating bone structural damage and reducing the level of bone resorption markers and the number of osteoclasts. The miR-27a inhibitor could counteract the protective effect of BMSC-EVs in OP mice. Further experiments showed that silencing Dkk2 reversed the stimulatory effects of the miR-27a inhibitor on OB and OC, resulting in the same protective effect as BMSC-EVs [66], indicating that the potential mechanism is that miR-27a activates Wnt/β-catenin signaling by targeting Dkk2, which ultimately promotes bone formation and inhibits bone resorption. The above studies showed that miR-27a released by BMSC-EVs regulates bone metabolism through Dkk2/Wnt/β-catenin signaling and plays a role in bone protection. It may serve as a potential target for the regulation of bone diseases. Wnt/β-catenin signaling can not only be inhibited by the Dkk family but also be regulated by Wnt inhibitor-1 (WIF-1) [99]. WIF1, a recombinant human Wnt inhibitor, directly interacts with various Wnt ligands to attenuate their binding to membrane-bound receptors. The epigenetic promoter methylation of WIF1 results in transcriptional silencing and Wnt signaling upregulation [100]. Wei et al. [89] found that miR-424-5p was enriched in OP patient-derived BMSC-Exos, while WIF1 was lowly expressed, which could significantly inhibit the osteogenic differentiation of BMSCs. Functional validation indicated that BMSC-Exos from OP patients might inhibit osteogenic differentiation through miR-424-5p/WIF1/Wnt/β-catenin.

The above studies suggest that multiple miRNAs enriched in BMSC-EVs may regulate osteogenic differentiation through Wnt/β-catenin signaling (as shown in Figure 1). By modifying specific miRNAs in BMSC-EVs, Wnt/β-catenin can be effectively activated to promote bone formation. Notably, Wnt/β-catenin signaling not only promotes bone formation but also negatively regulates osteoclastogenesis by promoting osteoprotegerin (OPG) secretion [101]. The interactions between monocytes-macrophages-osteoclasts and BMSC-osteoblasts play a crucial role in the regulation of bone metabolic diseases [102]. Therefore, future studies could focus on whether BMSC-EV-miRNAs can simultaneously target bone formation and bone resorption to regulate bone metabolic homeostasis.

### 3.3. BMP/Smad Signaling Mediating the Regulation of BMSC-EV-miRNAs on Bone Metabolism

Transforming growth factor-beta (TGF-β)/bone morphogenic protein (BMP) signaling is involved in the regulation of most cellular processes, especially in osteoblast differentiation and bone formation [103]. Studies have shown that TGF-β1 significantly promotes ectopic bone formation induced by BMP-2 [104]. BMP and activin membrane-bound inhibitor homolog (BAMBI) has a structure similar to TGF-β type I receptors and can react with BMP ligands to antagonize the TGF-β/BMP signaling pathway by inhibiting the formation of active ligand-receptor complexes, thereby indirectly inhibiting the effect of BMP ligands [72]. BMP ligands (such as BMP-2) first bind to the extracellular domains of type II receptors (BMPR2, ACVR2A and ACVR2B) to form dimers, which then phosphorylate type I receptors (BMPR2, ACVR2A and ACVR2B), and, finally, phosphorylation activates canonical Smad pathway proteins (Smad1/5/9) [105]. Therefore, BMP-2 is considered to be a key regulator of Smad signaling [106]. Bioinformatics and luciferase reporter assays revealed that BAMBI is a direct target gene of miR-20a [59]. Previous studies have shown that miR-20a may regulate BMP signaling and promote osteogenic mineralization by targeting the BMP-2 transcript [107,108]. After the co-culture of hBMSCs with BMSC-EV-miR-20a, BAMBI gene and protein expression were significantly downregulated, while the osteogenesis-related genes ALP, Runx2 and OCN were significantly upregulated, thereby promoting the osteogenic differentiation of hBMSCs. The overexpression of BAMBI partially reversed the osteogenic differentiation-promoting effect of BMSC-EV-miR-20a on hBMSCs [59]. This indicates that BMSC-EV-miR-20a may promote the osteogenic differentiation of hBMSCs by directly targeting BAMBI. However, this study did not further detect the expression changes of BMP/Smad signaling-related genes and proteins.

Studies have revealed that changes in the cellular microenvironment affect the type and content of cargoes enriched in exosomes [61,109]. The cargo components contained in exosomes secreted by BMSCs from patients with OP are significantly different from those from healthy human BMSCs [9]. Jiang et al. [86] found that, compared with BMSC-Exo from healthy people, BMSCs derived from OP patients treated with BMSC-Exo had upregulated the expression of miRNA-21 and downregulated the mRNA of osteogenesis-related factors such as ALP and Runx2, while BMSCs osteogenic differentiation was significantly inhibited. The overexpression of the target gene Smad7 of miRNA-21 could partially reverse the inhibitory effect of OP-BMSC-Exos on osteogenesis [86]. The above results suggest that miRNA-21 enriched in OP-BMSC-Exos may directly target Smad7 through sponge function to inhibit BMSC osteogenic differentiation. In addition to the abnormal bone metabolism that can affect the miRNA expression profile of BMSC-Exos, studies also showed that the expression of miRNAs enriched in exosomes extracted from BMSCs cultured in a normal medium and BMSCs cultured in an osteogenic differentiation induction medium may also be different. With the prolongation of osteogenic differentiation time, the type and expression of miRNAs changed continuously [62]. Through miRNA sequencing, Fan et al. [36] found that the expression of miR-29a was significantly decreased in exosomes extracted from the BMSC osteogenic differentiation medium compared with exosomes extracted from the BMSC growth medium, suggesting that miR-29a may be involved in mediating the promotion of BMSC-Exo osteogenic differentiation. Further studies showed that inhibiting miR-29a expression in BMSC-Exos activated BMP/Smad signaling by upregulating its target gene BMPR1A, thereby promoting the osteogenic differentiation of BMSCs. In contrast, the overexpression of miRNA-29a reduced the expression of BMPR1A and pSmad1/5/8 [36]. Furthermore, Liu et al. [19] explored the combined action mechanism of multicomponent miRNAs in BMSC-Exos in an osteogenic differentiation induction medium. Previous studies have reported that BMPR2 and ACVR2B may compete in Smad signaling. BMP-2 has a higher affinity for the type II receptors ACVR2B and ACVR2A than BMPR2 [110] and is involved in activating the canonical Smad2/3 pathway opposite to the action of Smad1/5/9 [111]. The miRNA mimics of let-7a-5p, let-7c-5p, miR-328a-5p and miR-31a-5p were transfected into BMSCs, respectively, and exosomes were extracted to detect the changes of BMP/Smad signal expression. The results showed that all four miRNA-modified exosomes significantly upregulated BMPR2 expression. Let-7a-5p, let-7c-5p and miR-328a-5p significantly downregulated ACVR2B and ACVR1, while miR-31a-5p downregulated only ACVR1 [19]. Notably, BMSC-Exos had no significant effect on BMP-2 expression in vitro but significantly increased the protein level of BMP-2 in vivo. This may be attributed to the complex cellular environment. In addition, all four miRNAs significantly upregulated the expression of pSmad1/5/9 and inhibited the expression of Smad2 and pSmad2 [19]. It was suggested that these four upregulated miRNAs in BMSC-Exos could target ACVR2B/ACVR1 and regulate the competitive balance of BMPR2/ACVR2B on the BMPR-induced phosphorylation of Smad1/5/9 [19]. This indicates that these multicomponent miRNAs induce osteogenic differentiation by directly targeting ACVR2B/ACVR1 and regulating the competing balance of BMPR-associated Smad1/5/9 phosphorylation and ACVR-associated Smad2/3 phosphorylation.

In conclusion, BMSC-EVs may affect osteogenic differentiation by regulating BMP/Smad signaling through its multiple key miRNAs (as shown in Figure 2), such as miR-20a, miRNA-21 and miR-29a. It should be noted that BMSC-EVs from different sources may have different miRNA expression profiles or even have opposite regulatory effects. As mentioned above, BMSC-EVs from OP patients might inhibit osteogenic differentiation via miRNA-21, while BMSC-EVs from healthy people might promote osteogenic differentiation via miR-20a. In addition, since BMSC-EVs are enriched for a variety of miRNAs, there may be complex mutual regulation between different miRNAs, which means that a single modification of a certain miRNA may not be enough to produce satisfactory therapeutic effects. Further research is therefore needed to explore the specific mechanism of miRNAs enriched in BMSC-EVs regulating bone metabolism in order to guide the treatment of BMSC-EVs in bone metabolic diseases.

### 3.4. Angiogenesis Mediating the Regulation of BMSC-EV-miRNAs on Bone Metabolism

Bone is a specialized (i.e., mineralized) connective tissue that is highly vascularized. Blood vessels provide bone tissue with the oxygen, nutrients and growth factors needed for growth and development [112]. Coupled angiogenesis and osteogenesis are critical for bone tissue repair and regeneration, such as fracture healing [113]. Previous studies have shown that angiogenesis precedes osteogenesis in bone remodeling and repair [114]. Angiogenesis dysfunction disrupts bone growth and remodeling, leading to OP or poor fracture healing [112]. Numerous studies have reported that BMSC-Exos may maintain bone homeostasis by regulating angiogenesis. BMSC-Exos promotes nonunion healing by promoting angiogenesis and osteogenesis [115]. Another study demonstrated that BMSC-Exos could significantly stimulate bone regeneration and angiogenesis in an ovariectomized (OVX) rat model of the critical-sized calvarial defect [85]. However, the above studies did not further explore the specific mechanism of BMSC-Exos regulation of bone blood vessels.

Previous studies have focused on the regulation of miRNAs on angiogenesis [116]. Therefore, the abundant miRNAs contained in BMSC-Exos may be involved in mediating the regulation of BMSC-Exos on bone angiogenesis. Lu et al. [9] revealed that miRNA-29a is enriched in BMSC-Exos, which can be taken up by human umbilical vein endothelial cells (HUVECs) to promote the proliferation, migration and tube formation of HUVECs, thereby promoting angiogenesis. Vasohibin 1 (VASH1) was identified as a direct target of miR-29a, mediating the effects of BMSC-Exo-miR-29a on angiogenesis [9]. Further studies showed that overexpression of miR-29a derived from BMSC-Exos showed a strong ability to promote angiogenesis and osteogenesis, while the inhibition of miR-29a derived from BMSC-Exos showed the opposite effect [9]. The above findings suggest that miR-29a in BMSC-Exos promotes angiogenesis and osteogenic differentiation by targeting VASH1. A newly discovered vascular subtype, type H vessels, has been reported to recruit osteoprogenitor cells and combine osteogenesis with angiogenesis [117], which is considered a crucial event during bone remodeling and repair [118]. However, this study did not find any genes associated with H-type angiogenesis among the predicted target genes of miR-29a [9], which means that BMSC-Exo-miR-29a may regulate angiogenesis during osteogenesis through yet-to-be-discovered signal transduction. Another study found that miR-214-3p was significantly increased in BMSC-Exos from OVX mice but significantly decreased in the BMSC-Exos of OVX mice with the knee mechanical loading group by miRNA sequencing analysis [30]. Notably, loading of the knee joint significantly improved H-type vessel formation and microvascular volume near the growth plate in OVX mice, as well as increased bone mineral density and bone mineral content. BMSC-Exos overexpression of miR-214-3p from BMSC-Exos reduced the angiogenic potential [30]. This suggests that knee mechanical loading enhances the formation of H-type blood vessels by downregulating exosomal miR-214-3p to promote bone angiogenesis and prevent OVX-induced bone loss. Mechanical stress can directly activate mechanical signaling molecules, which play an important role in vascular injury and repair [119]. However, only 2 weeks of passive mechanical loading of the knee was applied in the above study. Based on our previous studies [120,121], moderate intensity treadmill exercise is strongly recommended as an active exercise that may have a stronger stimulatory effect on bone vascularization and osteogenesis than passive mechanical stress loading of the knee joint. It should be noted that the beneficial effects of exercise may take at least 3-4 weeks to become apparent. In addition, Wang et al. [30] found that the overexpression of BMSC-miR-214-3p may downregulate the expression of the vascular endothelial growth factor (VEGF) and inhibit the tube formation and cell migration of HUVECs. However, the inhibition of BMSC-miR-214-3p could reverse the above results. Previous studies have shown that VEGF signaling couples angiogenesis and bone remodeling by affecting the formation of H-type blood vessels [122]. As a chemotactic molecule, VEGF can recruit endothelial cells to bone tissue and regulate the differentiation of osteoblasts and osteoclasts [114]. Thus, the expression of VEGF, which is important for H-type angiogenesis, is regulated by miR-214-3p. H-type angiogenesis was proved to be associated with exosomal miR-214-3p in response to mechanical stress. However, the specific regulatory mechanism is not fully understood.

Oxygen concentration has been considered to be critical during the proliferation, differentiation and self-renewal of MSCs [123]. MSCs are usually exposed to normoxia when cultured in vitro, but MSCs are in a hypoxic environment in vivo [124]. Liu et al. [26] demonstrated that, compared with ucMSC-Exos cultured under normoxia, ucMSC-Exos cultured under hypoxia (Hypo-ucMSC-Exos) may promote the high expression of miR-126 in Hypo-ucMSC-Exos by activating hypoxia inducible factor-1α (HIF-1α); and then the high expression of miR-126 activates the Ras/Erk pathway by targeting the Sprouty-related, EVH1 domain-containing protein 1 (SPRED1), causing significant angiogenesis, proliferation and migration and thereby promoting fracture healing. Therefore, hypoxic preconditioning of MSCs can significantly enhance the biological function and activity of MSC-Exos, representing an efficient and promising approach for promoting bone angiogenesis. Furthermore, in addition to the ucMSC-Exos selected in the above study, future studies could further explore the regulatory mechanism of BMSC-Exos in bone angiogenesis under hypoxic conditions.

### 3.5. RNA Methylation Modification Mediating the Regulation of BMSC-EV-miRNAs on Bone Metabolism

According to the evidence, RNA could not only transmit the genetic information of DNA to proteins but also participate in the regulation of many biological processes through diverse post-transcriptional modification functions [125]. At the RNA level, more than 100 post-transcriptional modifications have been identified, while m6A methylation is one of the most common modifications in mammals that regulate post-transcriptional gene expression without changing the base sequence [126]. m6A methylation is mainly involved in many biological processes and diseases through three types of proteins: methyltransferases, methylation readers and demethylases [127]. Among them, m6A demethylases, such as Alpha-ketoglutarate-dependent dioxygenase FTO (FTO), can remove m6A methylated groups in RNA, which is the basis for the dynamic reversibility of methylation [127]. FTO, also known as fat mass and obesity-associated protein, is one of the key elements in regulating body weight and fat mass, mainly by affecting lipogenesis [128]. Previous studies have shown that the expression of FTO is upregulated in the bone marrow and is involved in the regulation of BMSC osteogenic differentiation in mouse models of aging and OP [129]. Furthermore, Zhang et al. [83] showed that FTO is a potential target of miR-22-3p by bioinformatics analysis. miR-22-3p was enriched in EVs from BMSCs [130]. In the OVX mouse model, miR-22-3p expression was suppressed, whereas FTO expression was elevated. BMSC-EV-miR-22-3p can directly inhibit the expression of FTO, leading to the accumulation of m6A on MYC transcripts and thereby reducing the stability of MYC mRNA and inhibiting the phosphatidylinositol 3-kinase and protein kinase B (PI3K/AKT) signaling pathway, ultimately increasing the ALP activity and matrix mineralization. The inhibition of BMSC-EV-miR-22-3p was able to partially reverse the above expression, with reduced ALP staining and activity and extracellular matrix mineralization [83]. It was shown that BMSC-EV-miR-22-3p can directly inhibit the demethylase FTO to inactivate the MYC/PI3K/AKT pathway, thereby enhancing osteogenic differentiation. Furthermore, the BMSC-EVs of OP patients exhibited downregulated miR-29b-3p compared to non-OP patients. The decreased expression of miR-29b-3p leads to the upregulation of histone demethylase lysine demethylase 5A (KDM5A) [87]. Studies have shown that KDMs are extensively involved in the regulation of MSC lineage specification [131]. KDM4B, the demethylase K9 of histone H3, is required for the osteogenic differentiation of MSCs by removing H3K9me3 at the Dlx5 promoter region [132]. Additionally, KDM5A, which is highly expressed in the BMSCs of osteoporotic mice, regulates bone formation by demethylating H3K4me3 in its promoter region [133]. Further study found that KDM5A was enriched in the promoter region of cytokine signaling 1 (SOCS1). KDM5A can suppress SOCS1 expression by regulating the demethylation of H3K4me3 and H3K27ac in the promoter region of SOCS1 and by suppressing chromatin configuration [134], which leads to the inhibition of SOCS1 loss on the nuclear translocation of the downstream target factor nuclear factor kappa-B (NF-κB) p65, ultimately inhibiting the osteogenic differentiation of BMSCs [135]. Zhang et al. [87] showed that BMSC-EV-miR-29b-3p blocked the SOCS1/NF-κB pathway by specifically targeting the demethylase KDM5A, thereby enhancing the osteogenic differentiation of BMSCs. Therefore, the upregulation of BMSC-EV-encapsulated miR-29b-3p could serve as an effective strategy to promote osteogenesis.

The modification of m6A methylation is dynamic and reversible and plays a key role in miRNA processing, mRNA export translation and splicing [136]. Studies on the regulation of bone metabolism by BMSC-EV contents through m6A methylation have been limited to exploring the interaction of miRNAs with demethylases. Future research could focus on exploring how methyltransferases and demethylases work in concert and how methylation readers function after recognizing RNA methylation. In addition, whether other contents of BMSC-EVs, such as mRNA, can also affect the function of target cells and regulate bone metabolism through methylation modification is also worth exploring.

### 3.6. Other Types of ncRNAs Mediating the Regulation of BMSC-EV-miRNAs on Bone Metabolism

In recent years, multiple studies have shown that exosomes selectively enrich non-coding RNAs (ncRNAs), such as miRNAs, long non-coding RNAs (lncRNAs) and circular RNAs (circRNAs), which are involved in various cellular processes through intercellular communication [22,137]. According to the endogenous RNA competition hypothesis, all RNA transcripts, including ncRNAs and mRNAs, interact by competing RNA binding sites [121]. Previous studies have demonstrated that both lncRNAs and circRNAs can inhibit miRNA expression by acting as miRNA sponges to compete with mRNA for miRNA response elements [138]. LncRNAs are RNAs with a length of more than 200 nt without protein-coding function [121]. Long non-coding RNA metastasis-associated lung adenocarcinoma transcript 1 (MALAT1), also known as nuclear-enriched transcript 2 (NEAT2), has been shown to regulate bone metabolism by competitively binding miRNAs [139]. A large number of differentially expressed miRNAs are involved in SATB2-induced early osteogenic differentiation through the BMP signaling pathway [140]. Bioinformatics analysis showed that miR-34c could target MALAT1 and SATB2 to regulate osteogenesis [54]. The inhibition of BMSC-Exo-miRNA-34c and the overexpression of MALAT1 or SATB2 effectively promote osteogenic activity and alleviate OP in OVX mice [54], indicating that MALAT1 enriched in BMSC-Exos might enhance osteogenesis and alleviate OP in OVX mice through the miR-34c/SATB2 signaling axis. In addition, Yang et al. [141] showed that MALAT1 can participate in the regulation of osteolysis through OPG/RANKL/VEGF signaling, suggesting that MALAT1 may inhibit bone resorption. The above findings suggest that MALAT1 in BMSC-Exos may not only promote bone formation but also inhibit bone resorption, although the specific regulatory mechanism needs to be further explored.

In addition to lncRNAs, circRNAs are also enriched in BMSC-Exos, which may be involved in the regulation of bone metabolism [22]. CircRNAs are non-coding RNAs produced by backsplicing and are characterized by covalently closed-loop structures without 5′ to 3′ polarity or a poly-A tail [142]. Similar to lncRNAs and miRNAs, circRNAs are considered to be key regulators of cellular function rather than “nonfunctional by-products of aberrant RNA splicing” [143]. CircRNAs are evolutionarily conserved and have multiple functions, such as regulating gene transcription, alternative splicing, translation, interacting with RNA-binding proteins (RBPs) and acting as miRNA sponges to inhibit their expression [144]. CircRNAs are abundant in exosomes, and more than a thousand exosomal circRNAs have been identified in human serum [145]. The expressions of hsa_circ_0002060 and hsa_circ_0001275 in the blood of patients with OP were significantly upregulated with high sensitivity and specificity. ROC analysis indicated that hsa_circ_0002060 and hsa_circ_0001275 may serve as potential diagnostic biomarkers for postmenopausal OP [146,147]. In addition, Zhi et al. [137] analyzed the circRNA expression profiles of exosomes isolated from the serum of OP patients and healthy people by circRNA microarray technology. The results showed that hsa_circ_0006859 was most significantly upregulated in the blood exosomes of postmenopausal women with OP compared with the controls. The overexpression of hsa_circ_0006859 impeded osteoblast differentiation and promoted the adipogenic differentiation of hBMSCs by inhibiting miR-431-5p and upregulating Rho associated coiled-coil containing protein kinase 1 (ROCK1) (a target gene of miR-431-5p). The knockdown of ROCK1 partially reversed these results [137]. This indicates that BMSC-Exo-hsa_circ_0006859 can regulate the balance between osteogenic and adipogenic differentiation through the miR-431-5p/ROCK1 axis. In addition, other studies have found that BMSC-Exo-circ-Rtn4 regulates osteoblast proliferation and apoptosis [148]. Tumor necrosis factor-α (TNF-α), as a multifunctional cytokine, can not only act as an inducer of osteoblast apoptosis to inhibit bone formation but also act on osteoclasts by stimulating osteoblasts to produce cytokines such as interleukin-1 (IL-1) and IL-6, thereby promoting bone resorption and leading to bone loss [149]. Cao et al. [148] found that TNF-α increased miR-146a expression, inhibited cell viability and promoted apoptosis in a dose-dependent manner in MC3T3-E1 cells. Subsequently, after co-culturing MC3T3-E1 cells with exosomes derived from circ-Rtn4-modified BMSC-Exos, it was found that circ-Rtn4-modified BMSC-Exos inhibited the expression of miR-146a through sponge function and attenuated TNF-α-induced osteoblast cytotoxicity and apoptosis [148]. However, only cell lines were used in this study, and no primary osteoblasts, osteoclasts or animal experiments were performed to further validate the results of this study.

The above research results show that both lncRNAs and circRNAs enriched in BMSC-Exos can act as intracellular miRNA sponges or compete with miRNAs to participate in the regulation of downstream genes and encoded proteins, thereby affecting the proliferation and differentiation of osteoblasts. This suggests that modifying key lncRNAs or circRNAs in BMSC-EVs is a feasible and effective strategy for the treatment of bone metabolic diseases. However, there are few relevant research studies for reference, and further research is thus needed to explore the specific regulatory mechanism of BMSC-EV-enriched ncRNAs, especially of lncRNAs and circRNAs, on bone metabolism in order to guide the precise treatment of bone metabolic diseases.

## 4. Protein-Modified BMSC-EVs Regulating Bone Metabolism

Recent studies have emphasized that, in addition to ncRNAs, a large number of mRNAs and functional structural proteins are enriched in exosomes [150,151], which suggests that BMSC-Exos may also participate in the regulation of bone metabolism through their encapsulated mRNA and protein components [68]. Scleraxis (Scx) is a helix-loop-helix transcription factor, and the overexpression of Scx promotes bone formation [152,153]. Wang et al. [68] found that Scx is dynamically expressed in BMSCs during early tendon-bone healing. The local injection of exosomes derived from Scx-overexpressing BMSCs (BMSC-Scx) hindered osteoclastogenesis and inhibited abnormally activated peritunel osteolysis at the tendon-bone healing interface in a dose-dependent manner [68]. It is suggested that BMSC-Scx-Exos may be involved in regulating the function of osteoclasts, thereby affecting bone resorption. In addition, as previously mentioned, multiple miRNAs from BMSC-Exos regulate bone metabolism through the Wnt/β-catenin signaling pathway [88,89]. In the canonical Wnt/β-catenin signaling pathway, the activation of the Wnt protein promotes the nuclear translocation of the downstream crucial protein β-catenin, thereby initiating a series of downstream signaling cascades to enhance the osteogenic differentiation of BMSCs [154]. Glycoprotein non-melanoma clone B (GPNMB), also known as osteoactivin, is a multi-functional transmembrane glycoprotein expressed in various tissues including bone [155]. Huang et al. [156] emphasized that BMSC-EVs could significantly upregulate the expressions of Wnt1 and β-catenin and promote the osteogenic differentiation of BMSCs. Importantly, GPNMB-overexpressing BMSC-EVs (BMSC-GPNMB-EVs) exhibited a better effect on promoting osteogenic differentiation compared to control BMSC-EVs. The osteogenic differentiation induced by BMSC-GPNMB-EVs was partially inhibited by using the Wnt/β-catenin signaling pathway inhibitor Dkk1 [156]. This indicates that BMSC-GPNMB-EVs could promote the osteogenic differentiation of BMSCs through the Wnt/β-catenin signaling pathway. In addition to Wnt/β-catenin signaling, BMP/Smad signaling is also closely related to the regulation of bone formation and has been extensively studied. Noggin, a natural BMP antagonist, increases expression upon BMP stimulation as a negative feedback mechanism to prevent excessive BMP signaling [157]. Previous studies have shown that the overexpression of noggin inhibits osteogenic differentiation and bone formation [158], whereas the inhibition of endogenous noggin expression upregulates key osteogenic factors such as Runx2 and OCN by activating endogenous BMP/Smad signaling, thereby promoting bone formation [159]. Therefore, inhibiting the cognate binding protein noggin in BMSCs may upregulate the expression of BMP and promote the accumulation of osteogenesis-related factors in BMSCs, resulting in the enrichment of osteogenesis factors in BMSC-Exos. Fan et al. [36] reported that enhanced BMP signaling in exosomes from noggin-suppressed hMSCs could significantly promote bone regeneration. Interestingly, in this study, knocking down the noggin expression in hMSCs significantly reduced the expression of miR-29a, while the inhibition of miR-29a promoted the expression of its target gene BMPR1A to activate endogenous BMP/Smad signaling, which, together, promoted the pro-osteogenic effect of hMSC-Exos [36]. This suggests that noggin and miR-29a exhibited a synergistic effect of promoting osteogenic differentiation, but the potential cooperative mechanism was not clear. Therefore, the modified BMSC-Exos may exhibit complex endogenous cargo changes, leading to some unpredictable effects of BMSC-Exo-based therapeutic strategies.

In summary, the above studies preliminarily reveal that, by modifying functional proteins in BMSCs, the endogenous cargo of BMSC-EVs can be enriched to promote osteogenesis or inhibit osteoclastogenesis. This engineered BMSC-EVs exhibit powerful regulation of bone metabolism. However, the expression profile of the internal cargo of the modified BMSC-EVs may change drastically, suggesting that the protein-modified BMSC-EVs may exhibit some unpredictable effects. At present, there are few studies on the regulation of bone metabolism by protein-modified BMSC-EVs, and the underlying mechanism needs to be further explored. If appropriate modification targets are discovered, the cargo expression profile of BMSC-EVs can be precisely regulated, which will be an ideal therapeutic strategy for bone metabolic diseases.

## 5. Local and Systemic Administration of EVs

EV administration is widely recognized as an attractive cell-free approach for the management of bone metabolic diseases. Currently, the delivery of EVs includes carrier loading (local administration) mainly for the treatment of bone defects and intravenous injection (systemic administration) mainly for the treatment of OP.

### 5.1. Application of EVs in Bone Tissue Engineering for Local Administration

Unloaded EVs tend to be lost with fluid flow, which may limit the therapeutic efficacy of EVs [59]. Therefore, a carrier is required for local EV administration. An ideal carrier to load EVs for stable and sustained release may be a potential therapeutic strategy for bone regeneration and repair [19]. Due to the properties of good hydrophilicity, encapsulation ability, biocompatibility and biodegradability, biomedical hydrogels have been increasingly used as carriers to deliver cells, drugs or growth factors for engineered tissue repair [160]. In recent years, studies have revealed that encapsulated EVs in a modified hyaluronic acid hydrogel can be continuously released [161]. The porous structure of the hydrogel can release the encapsulated bioactive factors in a sustained manner. In particular, injectable biopolymer hydrogels are suitable for repairing bone defects, as they can be injected directly into the bone defect and provide a spatial fit between the implant and the randomly shaped defects [162]. Fan et al. [36] found that the injectable methacrylated glycol chitosan (MeGC) hydrogel with encapsulated BMSC-Exos could significantly promote bone repair in a critical-size bone defect model in mice. Although hydrogel encapsulation can provide stable sustained release as the material degrades, hydrogels are mainly applied in cartilage regeneration rather than bone regeneration due to their inherent mechanical properties. Therefore, a rational approach for bone regeneration is to incorporate scaffolds to fabricate composite biomaterials with high cytocompatibility and good mechanical properties [163]. Bioscaffolds, stem cells and growth factors are considered to be the three basic elements of bone regeneration and repair, i.e., stem cells or bioactive molecules (e.g., EVs) are loaded into biomaterial scaffolds for promoting bone regeneration and repair. Recently, the delivery of EVs on biomaterial scaffolds has been successively reported for bone regeneration and represents a promising strategy for bone defect repair. Qi et al. [85] reported that Exos loaded on β-tricalcium phosphate (β-TCP) bioscaffolds could significantly promote angiogenesis and bone regeneration in calvarial defect sites in OP rats. Further, Li et al. [164] developed a modified polydopamine-coated PLGA scaffold capable of the controlled release of Exos to effectively promote bone regeneration in mouse calvarial defects. Moreover, a hierarchical MBG scaffold with macro-/micro-/meso-porosities was developed [165]. The micron-scale porosity (0.5–2 μm) and high specific surface area of the hierarchical mesoporous bioactive glass (MBG) scaffold can load and shelter Exos, which is beneficial to protecting the possible damage to the phospholipid membrane structure of Exos during the freeze-drying process, thereby maintaining the biological activity of exosomes, and may be a potential carrier for improving exosome delivery in bone regeneration therapy [165]. Importantly, MBG scaffolds have inherent osteoinductive properties and promote ALP activity [166]. Subsequently, Liu et al. [19] employed this layered MBG scaffold to load BMSC-Exos and found that Exos was rapidly released in the first week, followed by a steady slow-release rate until the release percentage at day 28 was 75.42%. Furthermore, the BMSC-Exo-loaded MBG scaffolds were shown to rapidly initiate bone regeneration and promote bone mineral deposition to repair critical-sized calvarial defects in rats [19]. Therefore, the lyophilized delivery of BMSC-Exos on the hierarchical MBG scaffold enables the controlled release and maintenance of biomolecular activity by entrapment in the surface microporosity of the scaffold, thereby inducing a rapid initiation of bone regeneration. Since 3D printing technology can customize the shape and size of the material to match the bone defect [167], some scholars have attempted to encapsulate Exos in injectable hyaluronic acid hydrogel (HA-Gel) and fill the encapsulation into the pores of custom 3D printed nanohydroxyapatite/poly-ε-caprolactone (nHP) scaffolds [52]. The findings demonstrate that HA-Gel possesses properties of the long-term sustained release of exosomes, and the novel composite 3D-printed bioscaffold significantly enhanced angiogenesis and bone regeneration in a rat critical-size calvarial defect model [52]. Despite the possible engineering complexity and high cost, the high-efficiency bone regeneration demonstrated by this bone tissue engineering repair strategy is still exciting [168].

In summary, hydrogel can be used as an effective carrier for EVs for the local controlled release of EVs, and bioscaffolds can provide sufficient mechanical support for the repair of bone defects. Filling the bioscaffolds with EVs loaded in hydrogels can effectively enhance bone formation and angiogenesis, thereby promoting bone defect repair. However, it has to be admitted that, although this composite scaffold can effectively promote bone regeneration, its clinical application is hindered by its complexity and mass production cost. In addition, the preservation, controlled release and long-term activity maintenance of EVs are also worthy of attention. Further research is needed to optimize the EVs delivery strategy.

### 5.2. Bone-Specific Targeted Modification of EVs for Systemic Administration

While the bone tissue engineering of the local administration of BMSC-EVs has become a current research hotspot, less research has focused on the study of the systemic administration of BMSC-Exos for the treatment of systemic bone metabolic diseases, such as OP. Recent studies have found that BMSC-Exos can effectively enhance the osteoblast differentiation of BMSCs in vitro [9,86]. However, tail vein injection of BMSC-Exos cannot effectively increase the cortical bone mass and cannot effectively prevent OVX-induced postmenopausal OP [9,84]. The short circulation time of systemically administered exosomes hinders their targeting of diseased tissues. Furthermore, systemically administered exosomes rapidly accumulate in the liver and spleen but barely accumulate in bone tissue [169]. This may be part of the reason for the poor specific targeting performance of systemically administered BMSC-Exos to bone tissue. Therefore, it is necessary to develop a delivery strategy that can confer specific recognition and targeting of BMSC-Exos to bone tissue, making the systemic administration of BMSC-Exos more precise and efficient. Alendronate (Ale), the most widely used anti-OP drug in clinical administration, contains a P-C-P group of the pyrophosphate analog that specifically targets bone through its high affinity with hydroxyapatite crystals [170]. Wang et al. [171] used an azide (N3) group-modified Ale molecule and an alkynyl (DBCO) group-modified MSC-EVs to construct an Ale-MSC-EVs complex coupled by the chemoselective coupling method “click chemistry”. “Click chemistry” has the advantages of mild reaction, simple operation, easy purification of products and no harmful by-products. It is commonly used in cell markers and biomedical synthesis [172,173]. Wang et al. [171] showed that the bone-targeting ability of Ale-MSC-EVs was significantly improved and effectively prevented osteoporotic bone loss in vivo, indicating that Ale-MSC-EVs may be a new potential therapy for OP. It is suggested that the covalent binding modification by “click chemistry” can significantly improve the bone-specific targeting ability of EVs. In addition, by the same method of group modification, some scholars have prepared aptamer functionalized BMSC-Exos (BMSC-Exo-Apt) by conjugating single-stranded DNA/RNA oligonucleotides with BMSC-Exos [174]. The 5’ end of the aptamer can be modified with an aldehyde group to form a stable Schiff base by reacting with the amino group of the BMSC-Exos membrane protein. This specific aptamer was able to significantly promote the internalization of BMSC-Exos into BMSCs in vitro [174]. Luo et al. [84] further explored whether this BMSC-Exo-Aptamer complex could promote the specific accumulation of exosomes in bone tissue to promote bone regeneration and repair. After the weekly tail vein injection of BMSC-Exo-Apt for two months, compared with unmodified BMSC-Exos, BMSC-Exo-Apt treatment was able to significantly promote the bone mass and bone mineral density of the femur in both OVX and femoral fracture mouse models [84]. This shows that BMSC-targeting aptamer functionalized BMSC-Exos can target bone tissue to avoid rapid metabolism and clearance, thereby regulating bone formation. The BMSC-Exo-Aptamer may represent a new and effective strategy to specifically target bone formation. In addition to the several covalent group modifications mentioned above, the electroporation and membrane-anchored modification methods have also been applied to develop novel exosome-related drug delivery systems [175]. Cui et al. [168] constructed a drug delivery system based on engineered exosomes. The bone-targeting peptides modified with diacyl lipid tails were anchored on the surface of exosome membranes through hydrophobic interactions, and then siRNA of the specific gene Shn3 was loaded onto exosomes by electroporation. The exosomes were conjugated with the peptide SDSSD (Ser, Asp, Ser, Ser, Asp) through a diacyl lipid insertion method, which conferred the BMSC-Exo-siShn3 specific delivery of siRNA to osteoblasts, thereby mediating osteoblast Shn3 gene silencing. The systemic administration of BMSC-Exo-siShn3 to mice can simultaneously enhance osteogenic differentiation, promote H-type angiogenesis and inhibit osteoclast formation [168], indicating that this bone-targeted engineered exosome delivery system is a feasible and efficient cell-free therapy for the treatment of bone metabolic diseases, which also provides new insights for the targeted therapy of other diseases.

Taken together, unmodified BMSC-EVs show limited therapeutic efficiency. The lack of specific bone-targeting properties of native BMSC-EVs largely limits their use as an effective drug strategy. Appropriately modified engineered BMSC-EVs can efficiently target bone tissue to enhance osteogenesis. Although it is still in its infancy, group-modified engineered exosomes have shown great potential as an efficient bone target delivery system for regulating bone formation and bone resorption, which presents a new strategy for the systemic administration of metabolic bone diseases.

## 6. Conclusions and Perspectives

BMSCs are recognized as the most promising cells in bone regenerative medicine. However, the procedure for obtaining BMSCs is invasive, and stem cell therapy may suffer from low engraftment efficiency, immune rejection and potential ethical concerns. Therefore, BMSC-EVs, as an important mediator of the paracrine effect of BMSCs, have received more and more attention in the field of bone disease treatment. Specifically, modified engineered BMSC-EVs can target bone tissue and deliver their endogenous bioactive molecules to regulate uncoupled osteogenesis and bone resorption through local or systemic administration. In addition, it is also necessary to further explore the effect of BMSC- EVs on osteocytes, as they orchestrate osteoblast and osteoclast activity in relation to the actual metabolic/mechanical bone requirements [176]. The precise bone metabolism regulation in vivo is more and more complicated with respect to the cell culture systems, where only the co-culture of the different bone cells (and the effect of EVs on them) could clarify their interactions. The precise regulation of bone metabolism by BMSC-EVs is an attractive topic for future research. The research on the regulation of bone metabolism by BMSC-EVs is still in the preliminary stage, and the underlying mechanism is still unclear. Most of the current research focuses on the regulation of bone metabolism by miRNAs encapsulated in BMSC-Evs. The regulatory role of the numerous mRNAs, proteins and lipids contained in the BMSC-Evs cargo on bone metabolism has not been fully investigated.

Although BMSC-EVs have shown great potential in bone metabolic diseases, there are still a series of problems to be further elucidated, such as purification, mass production, identification, transportation, the preservation of exosomes and their half-life and long-term safety in vivo. One of the most important issues is that the administration concentration and dosage of exosomes have not been unified [177]. Regrettably, EVs at different concentrations were not used to explore the dose-effect of exosomes, and even the administration concentration and dose of exosomes were not described in detail in most of the experiments. Different doses of BMSC-EVs may have differences in the regulation of bone metabolism, which is a concern. Secondly, the source of the parental cells of exosomes is also of great significance, since the expression profile of the intrinsic cargo encapsulated in EVs depends on the state and microenvironment of their parental cells [178]. For example, cell sources of different species such as hBMSCs and mouse BMSCs (mBMSCs) may lead to differences in EVs cargo content, and BMSC-EVs extracted from mice of different ages may also have different therapeutic effects due to the decline of the stemness of BMSCs with aging or cell passage [91,92]. Furthermore, although it is possible to modify the endogenous cargo of BMSC-EVs, the optimal conditioning regimen to enhance its therapeutic potential is largely unknown due to the complex interaction mechanism between cargoes, such as the reciprocal targeted inhibition of miRNAs and mRNAs and possibly various epigenetic modifications [109].

## Figures and Tables

**Figure 1 pharmaceutics-14-01012-f001:**
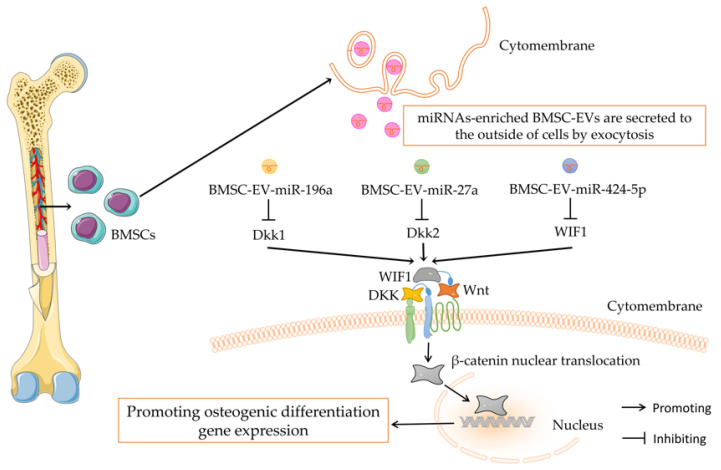
Wnt/β-catenin signaling mediates the regulation of BMSC-EV-miRNAs on bone metabolism.

**Figure 2 pharmaceutics-14-01012-f002:**
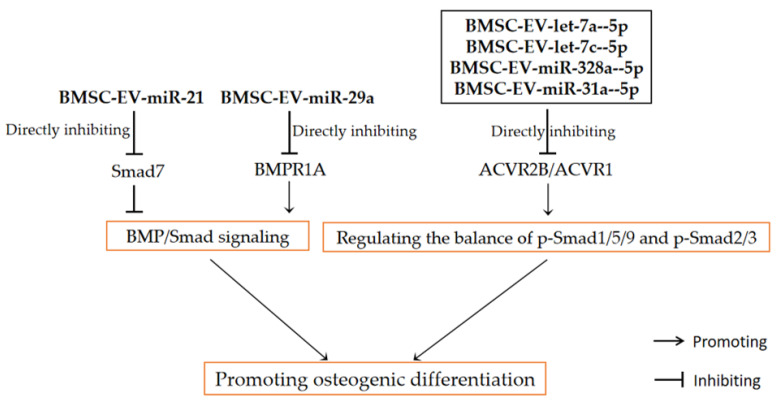
BMP/Smad signaling mediates the regulation of BMSC-EV-miRNAs on bone metabolism.

**Table 1 pharmaceutics-14-01012-t001:** BMSC-EV-miRNAs regulate bone metabolism.

Authors	miRNAs	Cell Source	Target Gene	The Function of miRNAs
Liu et al. [59]	miR-20a	BMSCs	BAMBI	Promoting osteogenesis
Jiang et al. [86]	miR-21	BMSCs	Smad7	Inhibiting osteogenesis
Zhang et al. [83]	miR-22-3p	BMSCs	FTO	Promoting osteogenesis
Wang et al. [66]	miR-27a	BMSCs	Dkk2	Promoting osteogenesis and inhibiting osteoclastogenesis
Luo et al. [84]	miR-26a	BMSCs	/	Promoting osteogenesis
Lu et al. [9]	miR-29a	BMSCs	VASH1	Promoting osteogenesis and angiogenesis
Zhang et al. [87]	miR-29b-3p	BMSCs	KDM5A	Promoting osteogenesis
Li et al. [11]	miR-101	BMSCs	FBXW7	Promoting osteogenesis
Qiu et al. [10]	miR-150-3p	BMSCs	/	Promoting osteogenesis
Li et al. [65]	miR-186	BMSCs	Mob1	Promoting osteogenesis
Peng et al. [88]	miR-196a	BMSCs	Dkk1	Promoting osteogenesis
Wang et al. [30]	miR-214-3	BMSCs	/	Inhibiting osteogenesis and angiogenesis
Wei et al. [89]	miR-424-5p	BMSCs	WIF1	Inhibiting osteogenesis
Zhang et al. [80]	miRNA-935	BMSCs	STAT1	Promoting osteogenesis
Wang et al. [68]	miR-6924-5p	BMSCs	OCSTAMP/CXCL12	Inhibiting osteoclastogenesis

## Data Availability

Not applicable.

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
