# Peer review of "Effects of BMSC-Derived EVs on Bone Metabolism"

_pharmaceutics, 2022, doi:10.3390/pharmaceutics14051012_

Round 1

Reviewer 1 Report

Major comments:

The review recapitulates recent literature mostly focused on BMSC-EV miRNA content as central players in the treatment of bone metabolic diseases. The paper is well presented, but, according the reviewer, some choices need to be discussed. One flaw of the review, for example, is that it completely ignores the existence and fundamental role of the osteocyte in orchestrating the activities of the bone cells that form and destroy bone.

Line 25-26 – The authors wrote: “Under normal conditions, osteoclast-dominated bone resorption and osteoblast-dominated bone formation are in a dynamic balance”.

Many authors do not recognize that these phases (i.e. bone resorption and formation) are not "dominated" but EXECUTED by osteoclasts and osteoblasts, respectively, that undergo the role of the osteocyte, which is the ORCHESTRATOR of all bone processes (see a recent review on the condition/function of the osteocyte (eg. https://doi.org/10.3390/jfmk6010028). The sentence should be rephrased as follows: “Under normal conditions, bone resorption and bone formation are performed, respectively, by osteoclasts and osteoblasts, whose activity is orchestrated by the osteocyte, that maintains the other bone cells in a dynamic balance”.

Line 28 – The authors wrote: “During the process of bone remodeling, …”.

Again, it is suggested to write “During the process of bone remodeling, triggered/modulated by osteocytes” 

Lines 112-114: The authors wrote: “Subsequently, it was found that MSCs are widely distributed throughout the body, including bone marrow, adipose tissue and umbilical cord, …”

Interesting source of EV is also the amniotic fluid (https://doi.org/10.1002/biof.1576).

Lines 406-407: Authors wrote: “exercise is strongly recommended as an active exercise that may have a stronger stimulatory effect on bone vascularization and osteogenesis”.

Authors should also consider that osteocyte is the mechano-sensor of bone and that pulsing loading, higher than the physiological setpoints (see the Mechanostat Theory by Frostt), applied to bone upregulates osteocyte signalling towards the osteoblasts, to increase osteogenesis in response to overload.

Lines 506-508: Authors wrote: “MALAT1 in BMSC-Exos may not only promote bone formation but also inhibit bone resorption, although the specific regulatory mechanism needs to be further explored” AND Lines 598-600: Authors wrote: “by modifying functional proteins in BMSCs, the endogenous cargo of BMSC-Exos can be enriched to promote osteogenesis or inhibit osteoclastogenesis.

Once again, authors should be include in the review also the effects of BMSC-Exos on osteocytes (if they are known), since they orchestrate/trigger/modulate/stop osteoblast and osteoclast activity in relation to the actual metabolic/mechanical bone requirements (it is advisable to consult and quote for example DOI: 10.1152/physrev.00043.2020). This aspect should however be discussed or at least cited in the review.

Lines 685-080: Authors wrote: “The precise regulation of bone metabolism by BMSC-EVs is an attractive topic for future research. The research on the regulation of bone metabolism by BMSC-EVs is still in the preliminary stage, and the underlying mechanism is still unclear.”

The precise bone metabolism regulation in vivo is more and more complicated/sophisticated with respect the cell culture systems where only co-culture of the different bone cells (and the effect of EVs on them) could be clarify their interactions.

Minor comments:

The review is focused on the potential role of BMSC-EVs on the bone regeneration and repair comparing them with UC-MSC or ADSC. Most of the manuscript describes the role of miRNA, while only a short paragraph is related to the one of EV-proteins. The authors explain that “At present, there are few studies on the regulation of bone metabolism by protein-modified BMSC-Exos”. According the reviewer, some studies on the effect of protein content of BMSC-EV, not only protein-modified BMSC-EVS, should be added. Moreover, the comparison with the proteins, involved on bone repair, found in EVs derived from UC-MSC or ADSC could be useful to complete this part.

Figure 2 is too similar to Figure 1 and does not help in understanding the mechanism underpinned BMP/Smad signaling in the regulation of BMSC-EV-miRNAs on bone metabolism.

The authors state that <<The terms ‘EVs’ and ‘Exos’ are in fact not strictly differentiated, and both terms are used to refer vesicle species obtained by sequential centrifugation and filtration. Since the concept of ‘EVs’ includes ‘Exos’, in this article, we prefer to use ‘EVs’>>.

However, through the paper “Exos” term occurs many times. The authors should uniform the text.

Line 90: The authors wrote “The exosome purification technology used in the current study…”

What “current study”? The reference n. 35 or the previous one. Please, clarify.

Line 366: The authors wrote “Bone is a highly vascularized connective tissue”.

A bit approximate. Please, specify “Bone is a specialized (i.e. mineralized) connective tissue, highly vascularized”.

Some typos should be cleaned:

Line 27: “also known as bone marrow mesenchymal stromal cells”, maybe “bone marrow mesenchymal stem cells

Author Response

Dear Editors and Reviewers,

We are submitting our revised manuscript entitled “Effects of BMSC-derived EVs on bone metabolism” (pharmaceutics-1689116) to the Pharmaceutics for your reconsideration of its suitability for publication. All authors have read and approved the revised manuscript. We deeply appreciate the time and effort you have spent in reviewing our manuscript. We have learned much from the reviewers’ comments, which are valuable and constructive. After studying the comments and your advice carefully, we have made a corresponding revision. Words in red are the changes we have made in the new manuscript. Our responses to the comments are listed below:

Reviewer1

Major comments:

The review recapitulates recent literature mostly focused on BMSC-EV miRNA content as central players in the treatment of bone metabolic diseases. The paper is well presented, but, according the reviewer, some choices need to be discussed. One flaw of the review, for example, is that it completely ignores the existence and fundamental role of the osteocyte in orchestrating the activities of the bone cells that form and destroy bone.

Line 25-26 – The authors wrote: “Under normal conditions, osteoclast-dominated bone resorption and osteoblast-dominated bone formation are in a dynamic balance”.

Many authors do not recognize that these phases (i.e. bone resorption and formation) are not "dominated" but EXECUTED by osteoclasts and osteoblasts, respectively, that undergo the role of the osteocyte, which is the ORCHESTRATOR of all bone processes (see a recent review on the condition/function of the osteocyte (eg. https://doi.org/10.3390/jfmk6010028). The sentence should be rephrased as follows: “Under normal conditions, bone resorption and bone formation are performed, respectively, by osteoclasts and osteoblasts, whose activity is orchestrated by the osteocyte, that maintains the other bone cells in a dynamic balance”.

Reply: Many thanks to you for correcting our misunderstandings. We carefully read the recommended literature, which is an important supplement to our knowledge of bone metabolism and will also help us improve our follow-up experimental research design. We have made corresponding changes in the text. Thanks again for your constructive comments.

Line 28 – The authors wrote: “During the process of bone remodeling, …”.

Again, it is suggested to write “During the process of bone remodeling, triggered/modulated by osteocytes”

Reply: Thanks for your suggestion. We have modified the sentence.

Lines 112-114: The authors wrote: “Subsequently, it was found that MSCs are widely distributed throughout the body, including bone marrow, adipose tissue and umbilical cord, …”

Interesting source of EV is also the amniotic fluid (https://doi.org/10.1002/biof.1576).

Reply: Thanks for your comments, we have rounded up the omissions and added relevant references.

Lines 406-407: Authors wrote: “exercise is strongly recommended as an active exercise that may have a stronger stimulatory effect on bone vascularization and osteogenesis”.

Authors should also consider that osteocyte is the mechano-sensor of bone and that pulsing loading, higher than the physiological setpoints (see the Mechanostat Theory by Frostt), applied to bone upregulates osteocyte signalling towards the osteoblasts, to increase osteogenesis in response to overload.

Reply: Thanks for your kind suggestion. In the cited literature, the authors found that angiogenesis can be enhanced by using passive motion of the knee joint. Based on the previous studies of our research group, active treadmill exercise in mice is highly recommended, which can better promote osteogenesis and angiogenesis, compared to passive exercise. We agree with your suggestion and have made appropriate changes in the manuscript.

Lines 506-508: Authors wrote: “MALAT1 in BMSC-Exos may not only promote bone formation but also inhibit bone resorption, although the specific regulatory mechanism needs to be further explored” AND Lines 598-600: Authors wrote: “by modifying functional proteins in BMSCs, the endogenous cargo of BMSC-Exos can be enriched to promote osteogenesis or inhibit osteoclastogenesis.

Once again, authors should be include in the review also the effects of BMSC-Exos on osteocytes (if they are known), since they orchestrate/trigger/modulate/stop osteoblast and osteoclast activity in relation to the actual metabolic/mechanical bone requirements (it is advisable to consult and quote for example DOI: 10.1152/physrev.00043.2020). This aspect should however be discussed or at least cited in the review.

Reply: The effects of BMSC-Exos on osteocytes is a very interesting and meaningful study, although there are no related studies yet. We think this is a very attractive research point. It is supplemented in the summary section of this review and the relevant literature is appropriately cited.

Lines 685-080: Authors wrote: “The precise regulation of bone metabolism by BMSC-EVs is an attractive topic for future research. The research on the regulation of bone metabolism by BMSC-EVs is still in the preliminary stage, and the underlying mechanism is still unclear.”

The precise bone metabolism regulation in vivo is more and more complicated/sophisticated with respect the cell culture systems where only co-culture of the different bone cells (and the effect of EVs on them) could be clarify their interactions.

Reply: Thanks for your suggestion, we have supplemented it in the summary section of the text.

Minor comments:

The review is focused on the potential role of BMSC-EVs on the bone regeneration and repair comparing them with UC-MSC or ADSC. Most of the manuscript describes the role of miRNA, while only a short paragraph is related to the one of EV-proteins. The authors explain that “At present, there are few studies on the regulation of bone metabolism by protein-modified BMSC-Exos”. According the reviewer, some studies on the effect of protein content of BMSC-EV, not only protein-modified BMSC-EVS, should be added. Moreover, the comparison with the proteins, involved on bone repair, found in EVs derived from UC-MSC or ADSC could be useful to complete this part.

Reply: Thank you very much for your kind suggestion. First, there are few studies on the effect of protein content of BMSC-EVs. After reviewing the literature again, to our best knowledge, we have not omitted relevant literature. In this review, protein-modified BMSC-EVs refers to enrichment or lack of a certain protein in BMSC-EVs by overexpressing or inhibiting gene expression in BMSCs. This is the conventional method to verify the effect of protein content of BMSC-EVs - functional verification. Therefore, in general, to explore the effect of protein content of BMSC-EVs, the methods of overexpression or gene repression are used to modify the proteins in BMSC-EVs. Thanks again for your suggestion.

Furthermore, this review does not focus on comparing BMSC-EVs with other types of MSC-derived EVs, only a brief overview of why this paper chooses to focus on BMSC-EVs rather than uc-MSC or ADSC at the end of the second section "Overview of EVs". Therefore, the main theme of this narrow review is to summarize the effects of BMSC-EVs on bone metabolism. If uc-MSC or ADSC-related content is added to the protein-modified BMSC-EVs section, it may not match the title and theme.

Figure 2 is too similar to Figure 1 and does not help in understanding the mechanism underpinned BMP/Smad signaling in the regulation of BMSC-EV-miRNAs on bone metabolism.

Reply: Thanks to your suggestion, we have modified Figure 2 appropriately. It is worth mentioning that Figure 1 involves complex physiological processes related to Wnt/β-catenin signaling, such as nuclear translocation. Therefore, the regulation mechanism of Wnt/β-catenin signaling in the regulation of BMSC-EV-miRNAs on bone metabolism can be clearly demonstrated in the Figure 1. However, the research related to BMP/Smad signaling is relatively superficial, and the underlying mechanism has not been further studied. The Figures in this review can only be drawn based on existing research. Therefore, after careful consideration, we modified Figure 2 to show the mechanism underpinned BMP/Smad signaling in the regulation of BMSC-EV-miRNAs on bone metabolism in the form of a flowchart.

The authors state that <<The terms ‘EVs’ and ‘Exos’ are in fact not strictly differentiated, and both terms are used to refer vesicle species obtained by sequential centrifugation and filtration. Since the concept of ‘EVs’ includes ‘Exos’, in this article, we prefer to use ‘EVs’>>.

However, through the paper “Exos” term occurs many times. The authors should uniform the text.

Reply: Since this article is a review, many research articles are cited. In the cited literature, "Exos" or "EVs" are used. To ensure that the research conclusions of the literature are not misunderstood, we did not rashly replace all "Exos" with "EVs" in the cited literature. But in the summary and commentary in this review, we prefer to use "EVs". Thanks for your careful review. We have again proofread the full text in detail and revised some terms.

Line 90: The authors wrote “The exosome purification technology used in the current study…”

What “current study”? The reference n. 35 or the previous one. Please, clarify.

Reply: Thank you for pointing out the ambiguity here. Our original intention was to express that in most of the current studies, the purification of exosomes mainly relies on the detection of exosome diameter and surface molecular markers. We have revised this ambiguous sentence in the text.

Line 366: The authors wrote “Bone is a highly vascularized connective tissue”.

A bit approximate. Please, specify “Bone is a specialized (i.e. mineralized) connective tissue, highly vascularized”.

Reply: Thanks for your suggestion, we have made the corresponding changes based on your comments.

Some typos should be cleaned:

Line 27: “also known as bone marrow mesenchymal stromal cells”, maybe “bone marrow mesenchymal stem cells

Reply: Thanks for your careful review, we have fixed the typo.

We hope the Reviewers will be satisfied with the revisions for our manuscript. If you have any questions about our review, please do not hesitate to contact us.

Thanks and Best regards!

Yours Sincerely,

Prof. Guoxin Ni

05/03/2022

Reviewer 2 Report

The manuscript reviews the various functions of bone marrow mesenchymal stem cell (BMSC) derived extracellular vesicles (EVs) in bone formation and resorption. The review is well written, thorough with up to date literature and, most importantly, offers a nuanced perspective on the applications of BMSC derived EVs for therapeutic applications. However, there are a few caveats that may be addressed.

  1. The sections pertaining to the application of EVs for treating bone metabolic diseases could be organized to cover challenges associated with isolation and enrichment of EVs.
  2. Moreover, while considerable attention has been given to the methods for targeting EV based therapeutics to bone, the authors may consider going into greater detail on the use of scaffolds and hydrogels for site specific delivery and spatiotemporal control over release.

Author Response

Dear Editors and Reviewers,

We are submitting our revised manuscript entitled “Effects of BMSC-derived EVs on bone metabolism” (pharmaceutics-1689116) to the Pharmaceutics for your reconsideration of its suitability for publication. All authors have read and approved the revised manuscript. We deeply appreciate the time and effort you have spent in reviewing our manuscript. We have learned much from the reviewers’ comments, which are valuable and constructive. After studying the comments and your advice carefully, we have made a corresponding revision. Words in red are the changes we have made in the new manuscript. Our responses to the comments are listed below:

Reviewer2

  1. The sections pertaining to the application of EVs for treating bone metabolic diseases could be organized to cover challenges associated with isolation and enrichment of EVs.

Reply: Thanks for your constructive suggestion. By reviewing the literature, combined with the next revision suggestion, we supplement the description of issues such as EV extraction and preservation in Section V and the Summary section. Thanks again for your suggestion.

  1. Moreover, while considerable attention has been given to the methods for targeting EV based therapeutics to bone, the authors may consider going into greater detail on the use of scaffolds and hydrogels for site specific delivery and spatiotemporal control over release.

Reply: Thanks for your comments. After reviewing the literature, the related content of scaffolds and hydrogels (5.1. Application of EVs in Bone Tissue Engineering for local administration) is supplemented in the section of “Local and systemic administration of EVs”. We believe that this meaningful revision will make our article completer and more rigorous.

We hope the Reviewers will be satisfied with the revisions for our manuscript. If you have any questions about our review, please do not hesitate to contact us.

Thanks and Best regards!

Yours Sincerely,

Prof. Guoxin Ni

05/03/2022

Reviewer 3 Report

This review summarized the specific mechanism of BMSC-EVs regulating bone, and provided a basis for future research of BMSC-EVs in bone metabolic diseases. Furthermore, the authors extensively discussed the raised issues of BMSC-EVs, including purification, mass production, identification, transportation, preservation of exosomes and their half-life and long-term safety in vivo, and suggested a way to overcome these limitations. In particular, the review about bone-specific targeted modification of BMSC-EVs for systemic administration was very interesting. This is an informative, well-written, balanced, and timely review on a subject of considerable current interest. I have no additional comments about this paper. 

Author Response

Thank you very much for your endorsement of our review. We deeply appreciate the time and effort you have spent in reviewing our manuscript.

Round 2

Reviewer 1 Report

The authors responded fully to all of the reviewer's requests and modified the original text accordingly. the integration of the literature was also appreciated. The manuscript is improved and, in the reviewer's opinion, can be published in the current version.

 Final opinion of the Reviewer: the manuscript is acceptable in the present form for publication.